# Self-Reported Cognitive Aging and Well-Being among Older Middle Eastern/Arab American Immigrants during the COVID-19 Pandemic

**DOI:** 10.3390/ijerph20115918

**Published:** 2023-05-23

**Authors:** Linda Sayed, Mohammed Alanazi, Kristine J. Ajrouch

**Affiliations:** 1James Madison College, Michigan State University, 842 Chestnut Rd, East Lansing, MI 48825, USA; 2College of Nursing, Michigan State University, East Lansing, MI 48825, USA; alanazi4@msu.edu; 3Department of Nursing, College of Applied Medical Sciences, University of Bisha, Bisha 67714, Saudi Arabia; 4Department of Sociology, Anthropology and Criminology, Eastern Michigan University, Ypsilanti, MI 48197, USA; kajrouch@umich.edu; 5Institute for Social Research, University of Michigan, Ann Arbor, MI 48109, USA

**Keywords:** depression, mental health, immigration, support systems, older adults, Middle Eastern/Arab Americans

## Abstract

Background: The COVID-19 pandemic posed new challenges for cognitive aging since it brought interruptions in family relations for older adults in immigrant communities. This study examines the consequences of COVID-19 for the familial and social support systems of aging Middle Eastern/Arab immigrants in Michigan, the largest concentration in the United States. We conducted six focus groups with 45 participants aged 60 and older to explore participant descriptions of changes and difficulties faced during the pandemic relating to their cognitive health, familial and social support systems, and medical care. The findings indicate challenges around social distancing for older Middle Eastern/Arab American immigrants, which generated three overarching themes: fear, mental health, and social relationships. These themes provide unique insights into the lived experiences of older Middle Eastern/Arab American adults during the pandemic and bring to light culturally embedded risks to cognitive health and well-being. A focus on the well-being of older Middle Eastern/Arab American immigrants during COVID-19 advances understanding of how environmental contexts inform immigrant health disparities and the sociocultural factors that shape minority aging.

## 1. Introduction

The COVID-19 pandemic presented new challenges for older adults and generated potential risks to their cognitive aging. These challenges were particularly noteworthy due to interruptions in family relations, particularly for older adults in immigrant communities. Social isolation and older age are both factors associated with cognitive decline [1]. Socialization restrictions were important to preventing the spread of COVID-19, especially for older adults; however, they could have negative consequences for cognitive and mental health [2]. These restrictions limited the time spent with family and caregivers as well as reducing opportunities to engage in social and community activities, all of which can exacerbate cognitive deterioration [3,4]. Quarantine measures such as social distancing can lead to social isolation and loneliness that increases the risk of developing depression and anxiety [5,6,7,8,9,10,11,12]. Previous studies have shown that older adults in immigrant communities in the US were more likely to be disproportionately affected by the COVID-19 pandemic due to limited access to healthcare, living in poverty, and fear of legal consequences [13]. However, very few studies have shown the impact that COVID-19 had on Middle Eastern/Arab Americans older adults [14]. The experiences of the Middle Eastern/Arab American community during the COVID-19 pandemic provides a unique opportunity to understand how drastic changes to the support environment of older adults have a long-term impact on older adult well-being. Accordingly, this study was conducted to investigate the impact of COVID-19 on aging Middle Eastern/Arab immigrants in Michigan. Michigan is the home of the largest, most visible concentration of Middle Eastern/Arab Americans in the US [15].

Previous studies have shown that older adults experience an unavoidable biological process that affects every aspect of their lives. Growing older is associated with physical, psychological, and social changes [2,16]. Cognitive aging is defined as distinct changes in cognitive function including decline in memory, processing information, and executive function [17,18]. These detrimental changes negatively affect older adults’ well-being [19]. This decline contributes to health problems (e.g., dementia), disability, or loss of self-care capacity, and an increase in social care needs [2]. Social disconnectedness and isolation can increase the risk of mental health problems, which is a risk factor for incident dementia [5,9]. The imposed social isolation and reduction in everyday social contacts related to COVID-19 contributed to severe psychological symptoms among older adults [3,4]. These social support structures were limited during the pandemic shutdowns. During the first nine weeks of the pandemic, depression worsened in 54% of older adults living with dementia [5,9]. Depression is a risk factor for cognitive impairment, with depressed older adults having a 50% increased risk of developing dementia [20].

Middle Eastern/Arab Americans faced unique challenges during the COVID-19 pandemic. This is due to the reliance placed on family and friends for social support, and the cultural expectations placed on family members to care for older adult members [21,22,23]. Research has shown that Middle Eastern/Arab Americans report more contact frequency with their networks than Blacks and Whites [24]. Although social distancing measures were crucial in mitigating the impact of the COVID-19 pandemic, the interruption in familial and social relationships among Middle Eastern/Arab Americans left them with limited systems of support. Language barriers, food insecurity, and limited healthcare access made immigrant communities more vulnerable to contracting the virus and developing severe COVID-19 [13]. The aim of this study was to investigate how social distancing measures influenced older Middle Eastern/Arab American immigrants’ everyday lives and their well-being.

## 2. Materials and Methods

### 2.1. Study Design

This study was designed to collect key information about the impact of COVID-19 socialization restrictions on the familial and social support systems among aging Middle Eastern/Arab American immigrants. A qualitative approach was used as it contributes to a deeper understanding of lived experiences as it captures cultural norms and perspectives among participants [25]. Participants were asked to complete a short demographic questionnaire (Appendix A). The collected information is presented in Table 1. Semi-structured focus group interviews provide a quick and convenient technique to collect qualitative data from multiple people simultaneously, while also permitting an exchange of opinions on the subject. Additionally, focus group interviews utilize within-group discussion to validate perspectives and enhance credibility [26]. Focus group discussions were conducted in Arabic and English, based on participants’ linguistic preference, in Michigan between January and August 2022. 

### 2.2. Participants

A total of six focus groups were conducted with 45 participants aged 60 and older (mean age 73.6 ± 8.2 and 75.6% female). All the participants had immigrated to the United States and identified as Middle Eastern and/or Arab (Table 1). The participants in this study were from diverse Middle Eastern and Arab American communities in Michigan and were identified through local religious settings (i.e., churches and mosques) and other social organizations (i.e., NGOs) (Appendix A). Organizations were identified based on their engagement within communities of high concentration of Middle Eastern and Arab immigrants as well as their willingness to support this research study. All participants were given a $25 gratuity gift card for their participation.

### 2.3. Data Collection and Analysis

All the interviews were administered by L.S. and M.A. At the start of each focus group, the interviewers informed the participants of the objectives of the study, their rights to withdraw, the consent procedure, and the confidentiality and anonymity of the data collected. Participants’ consent and sociodemographic information were collected at the beginning of the interviews. Semi-structured interviews were conducted using a script designed according to the study objectives and a literature review (Table 2). The semi-structured interview was based on three open themes centered on the following areas: (1) general background and relationships, (2) medical care, and (3) sociopolitical background. Each focus group lasted between 60 and 90 min and took place in person at six different Middle Eastern and Arab organizations. We explored participant descriptions of the changes and difficulties they faced during the COVID-19 pandemic relating to their cognitive health, familial and social support systems, and medical care. We also explored the importance of immigrant sociopolitical experiences in managing the stress associated with the COVID-19 pandemic and the impact it had on their social relationships and lived experiences. All the interviews were audiotaped and transcribed verbatim, along with field notes to document group interactions, among other aspects. Inductive analysis was used to analyze and code the interviews using NVivo software (v. 1.6.2). NVivo was used to organize the analysis of the focus group transcripts, securely classify codes, and develop a codebook to identify themes across the groups.

### 2.4. Rigor

Two co-authors (LS and MA) validated the transcripts and translations. Interviews conducted in Arabic were translated to English by a co-author (MA) who is bilingual in Arabic and English. Translations were then validated by the first author (LS) who is fluent in both languages. Coding was performed independently by the aforementioned co-authors and then reviewed accordingly to achieve consensus.

### 2.5. Ethical Considerations

Prior to conducting this study, an Institutional Review Board (IRB) approval was obtained from Michigan State University (Approval No. STUDY00006535). All the participants in this study were provided with written informed consent information in Arabic or English based on their language preference. All audio recordings of the focus group interviews were kept in a secure, password-protected laptop during transportation and then transferred to secure cloud storage that had access restricted to only authorized members of the research team. All the participants were assigned a unique identification number to keep their identities anonymous.

## 3. Results

Throughout the focus groups, the participants discussed the impact of COVID-19 on their daily lives and the changes it generated. The focus of the conversations centered on changes to their familial and social support systems. Participants expressed feelings of social disconnect, loneliness, and fear during the pandemic. This study used an inductive content analysis method to analyze the qualitative data and discover emerging themes. Table 3 represents the frequency of the themes across the focus groups along with the description of each theme. A total of 3 major themes were identified including: fear, mental health, social relationships.

### 3.1. Fear

Fear involves the experience of being afraid, worried, or uncertain during the COVID-19 pandemic. Many participants shared their experiences, thoughts, and emotions related to COVID-19 fears. This theme emerged in all six of the focus groups. Phrases such as ‘terrified’ and ‘terror’ emerged in five focus groups. Participants compared their lives before and during the pandemic to describe the difficulties that they faced. The most common source of fear expressed by participants was related to the inability to see family members due to the COVID-19 restrictions. Participants reported concerns about the potential risk of contracting the virus, as well as feeling isolated and lonely, which contributed to their fears and increased their sense of vulnerability. The discussion (or narratives if not part of a focus group discussion) presented below illustrates sources of fear arising from the pandemic:**Participant #31 (female) FG 4**

“It was difficult for them [family] because I was alone. They wanted to be home because they were afraid of themselves and if something happened to me. It was hard, it was very hard. I think about myself that I’m a strong woman, I was very, very scared.”

This narrative illustrates the deep fear of getting COVID-19. While protecting older adults from potentially dying was the aim of restrictions, this resulted in a very acute awareness that they were alone and isolated. For our participants, these fears, which were triggered at the start of the pandemic, continued to linger, and perpetuated feelings of uneasiness and despair, particularly when engaging in social activities. These fears disrupted social engagement and altered the social environment for Middle Eastern/Arab American immigrants even after the restrictions were lifted. Through the usage of focus groups, our study allowed participants to engage in more open conversations about their personal experiences and lifestyle changes due to COVID-19. It was in these exchanges that participants expressed their common fears and worries as a result of the pandemic. For example:**Participant #35 (female), Participant #36 (female), Participant #38 and LS, FG 5**

P36: The worst impact was depression and fear. Fear was so bad, I was terrified.

LS: Fear of COVID or fear that something is gonna happen to you?

P36: To get sick and be a burden on my kids or get it to my kids, that was my biggest fear. 

P35: Not to bother our family if we get disabled or you know need help.

P38: Even without COVID you worry about being a burden.

Participants were not only worried about contracting the virus but worried over the increased likelihood of being a burden on their loved ones. Their fear triggered intense feelings of loneliness and stress over the idea of being left alone. For example:**Participant #2 (female) FG 1**

“Before, if they did not come to us, we would call them and ask them why they did not come to us. But now, no, we do not blame them, because we know that they are busy, and the time is a little difficult. I haven’t been going outside for two years now. I’m afraid and secondly, where to go, I’m afraid. Also, we have neither relatives nor friends. I don’t know what to say, they are also afraid, we are afraid of them, and they are afraid of us.”

Across the focus groups, participants acknowledged heightened fears produced by the pandemic that started with fear over their health and limited their mobility and interactions with their loved ones. These feelings led to self-imposed isolation, limiting their contact with extended family and friends. Such fears led older Middle Eastern/Arab American immigrants to take extra precautions in their relationships and limit their engagements outside of the home. Fear of contracting the virus could create poor well-being and generate additional health challenges such as limited physical activity and cognitive stimulation [27,28]. Due to these fears surrounding the virus, our participants expressed increased social disconnectedness and isolation, which has been linked to increased risks of depression and anxiety for older adults affecting their mental health [12,29].

### 3.2. Mental Health

Mental health issues include displaying signs of psychological distress, such as depressive symptoms, sadness, stress, and loneliness. Participants commonly reported having symptoms of depression, sadness, stress, and loneliness due to prolonged isolation and interruptions to their social life resulting from the pandemic. This had caused the participants to feel helpless and despair. Their discussion of their inability to engage in social activities or interact with their extended family and friends led to mental health concerns. Concerns over mental well-being have implications for cognitive health as heightened depression is linked to worsening cognitive function [12,29]. Fear over the COVID-19 virus is often intertwined with and triggers mental health vulnerabilities. These concerns were expressed among all the participants across the focus groups, and are exemplified in the narrative below.


**Participant #29 (female) FG 4**


“To answer your question, I specifically said I exercise almost at least five days a week. People who know me know what I do, physical activity, yoga, breathing techniques, everything by the book. But when I got COVID, I was, I stayed away from kids and grandkids. It was so difficult for me and sometimes I don’t want to get out of my bed. It’s a sign of depression. Sometimes, I stay only drinking water, sometimes I don’t wanna drink water, sometimes I drink four bottles of water and sometimes I go days without water. I used to walk around the house just to do my steps to compete with my grandkids; I didn’t have the strength to do that. I don’t like to cry unless I have to or there’s a reason. I found myself tearing, my eyes watering without no reason. And then I said OK, that’s it, if I don’t take care of myself no one will take care of me because I don’t want my kids around me for their safety not mine. My daughter used to come over and I will not open the door for her because they have kids, and we have a good relationship as mother and daughter. It just kills me to see them over the glass door. I didn’t want that and didn’t want them to come and be sad. They were sad and I was more sad because of them and their kids. I was scared of my life and was scared that I would die alone. I don’t wanna go to the hospital. I didn’t want to go to the hospital. So, it was a very scary moment, you don’t know what to do. You can sink deep down, or you can think wisely and move yourself from the hole you’re in.”

Focus groups allow participants to engage in more open conversations about their personal experiences and lifestyle changes due to COVID-19, while also permitting them to exchange various opinions on this matter. This also allowed for increased clarity on their well-being and mental health concerns by allowing for in-depth, open-ended discussions from the perspective of older Middle Eastern/Arab American immigrants. In these focus groups, participants identified what depression looked like and how they associate it with crying and feelings of madness. Participants also indicated that this resulted in less physical activity and engagement outside the home. Previous studies have shown the positive correlation between increased physical activity and a lower incidence of depression [30,31,32]. Depression among older adults is also associated with reductions in cognitive abilities and higher levels of suicide [33,34]. For older Middle Eastern/Arab American immigrants, experiences of depression coupled with fears over engaging in physical and social activity can negatively impact their cognitive well-being and mental health. Focus group participants elaborated the symptoms that led to their feelings of depression:**Participant #35 (female) and LS, FG 5**

P36: Depressed, I definitely got depressed.

LS: How did you know you’re depressed?

P36: You know how I got depressed, if anybody called me to check on me, I would start crying. I was depressed, I forced myself to go for a walk a little bit around the house but I was depressed.

Participants across the focus groups identified the psychological impact that the pandemic had on their well-being. They acknowledged the sadness, social isolation, and loneliness they felt due to the limited contact with their extended family and community. Participants identified being in a state of depression and being unable to leave their house despite socialization restrictions being lifted. In several exchanges, participants noted their need for psychological help despite the cultural hesitation and stigmatization surrounding mental health concerns [35], as shown in the following examples:**Participant #11 (male) FG 2**

“We want psychiatrists.” 


**Participant #45 (female) FG 6**


“I had a trauma from the whole world, and I had problems every day, fatigue at home, fatigue, not physical but psychological… My psyche suffered a lot, I got depressed and I don’t feel the taste of the world anymore… I felt internal destruction and I had depression.”

Across all six focus groups, participants indicated feelings of depression, anxiety, sadness, and stress related to the pandemic changes. Notably, they were able to identify the source of these feelings to social isolation, loneliness, and lack of family engagement. Participants acknowledged the change to their environment that left them feeling alone and distressed. These concerns over their well-being were linked to their social relationships and activities that were limited during the pandemic. Feelings of social disconnect were linked to expressions of isolation and depressive symptoms indicating the impact that social distancing had on their social relationships. 

### 3.3. Social Relationships

Social relationships involve changes in the ways older adults engage with immediate and extended family, and friends. Many participants reported feeling physically disconnected from their loved ones and their support system due to restrictions on family visits and social gatherings. Participants also reported feeling a sense of loss and grief due to the inability to spend time with loved ones, attend their funerals, or engage in social activities that once brought them joy. For example:**Participant#24 (female) FG 3**

“I have a few relatives and friends here. So, I would spend time at the mosque, gathering and social events and it would fill my time. All of that came to an end.”


**Participant#3 (female) FG 1**


“Life has changed and not like before, there are no occasions like before. When a person performs a wedding party, 500 people come to him, and currently 70, 80, 120, not more.”

Further, some immigrants who lost their family members during the pandemic were unable to attend their funerals, losing out on an important and meaningful ritual to mourn the loss of a loved one in the company of others. For Middle Eastern/Arab Americans, bereavement and grief over the passing of a loved one is not limited to the immediate family but is a community matter tied to particular cultural and religious traditions [36]. One participant expressed the following:**Participant#32 (female) FG 4**

“This happened with my father—they did not let us bury him, they took care of the burial. It was very difficult for us, they made us stay in the cars and did not allow us to come to him until they finished so that we could come to him and pray for him. It is very difficult—no matter what I say, I cannot describe what happened. It is very difficult for us. My sister and I, it was hard for us.”

Across the focus groups, participants acknowledged the disruptions to their social context and how they lost the communal connectedness they had previously relied on. Social spaces like churches and mosques were no longer available, which only heightened their feelings of loneliness and isolation. Being unable to perform religious and customary death rituals troubled our participants who saw these traditions disappearing. Even when things seemed to return to normal, interactions were limited and no longer generated the same community engagements. Many participants continued to fear leaving the house and became comfortable sitting at home alone with limited social interaction. For example:**Participant #20 (male) FG 3**

“I used to like to socialize with people, but I became withdrawn from people completely.” 


**Participant #40 (female) FG 5**


“It affected my social life; you don’t go out with your friends for a bite to the restaurant. I stopped going to exercise at the senior center. I stopped going to the movies—I used to go once a week. In church of course it’s fine but after church we used to go down and socialize with the congregation after the service. Now for a while nobody would come down. So, you lost track with a lot of your friends.”

Participants acknowledged the ways the pandemic had made them socially disconnected and had led to a loss of social connectivity within their communities. The potential long-terms effects of the pandemic is found in that despite the lifting of social restrictions, participants continued to be withdrawn and isolated from their social environment and loved ones. 

## 4. Discussion

The COVID-19 pandemic has had profound consequences for the mental health and well-being of older adults and understanding this impact is critical to address the challenges and implications for their well-being. This focus group study aimed to explore the consequences of COVID-19 for the well-being of aging Middle Eastern/Arab immigrants in Michigan. Specifically, this study sought to capture the lived experiences of older Middle Eastern/Arab immigrants through a focus group approach. By taking a qualitative approach, this study provides unique insights into the lived experiences of older Middle Eastern/Arab American adults and brings to light the concerns impacting their mental health and cognitive well-being. Such an approach allowed for the voices and perspectives of our participants to be exchanged in an open environment. 

Three key themes have been identified after qualitative analysis to illustrate the difficulties associated with the COVID-19 pandemic: (1) fear; (2) mental health issues; and (3) changes in social relationships. Our participants recognized the struggles of the pandemic that produced fears and worry. Across the groups, individuals described feelings of depression and sadness during the pandemic. Social distancing produced uncertainty about being alone and generated feelings of isolation and loneliness. Changes in the supportive environment and system for older Middle Eastern/Arab American immigrants generated heightened stress, anxiety, and loneliness, posing risks to their cognitive health and well-being. Research has shown that increased stress and social isolation has a negative impact on cognitive health and well-being for older adults [5,12]. For older Middle Eastern/Arab American adults in this study, pandemic restrictions to social support systems have increased mental health concerns and may increase the risk for prolonged psychological distress. Their concerns centered on their inability to connect with extended family and friends and uphold cultural and religious traditions, which resulted in feelings of fear, worry, and loneliness, indicating mental health effects on older Middle Eastern/Arab American immigrants. 

The findings of the current study align with the existing literature on the impact of COVID-19 pandemic on the well-being of older adults and its implication on their cognitive health [3,4]. However, in the case of older Middle Eastern/Arab American immigrants, older adults recognized that the loss of their support and communal environment negatively influenced their psychological well-being. For our participants, the fears generated during the pandemic were linked to heightened social disconnectedness, often self-imposed, and levels of depression. As noted in the focus group discussions, great difficulties going back to previous patterns and frequency of social interactions were experienced by our participants, revealing the ongoing challenges they face. These concerns were intimately and centrally intertwined with the breakdown of social relationships and culturally informed support structures. 

### 4.1. Implications for Research 

The findings of the present study have crucial implications for immigrant communities. Using older Middle Eastern/Arab Americans during the COVID-19 pandemic as a case study, the findings show the emphasized challenges around social distancing that limited contact with family and friends. The breakdown of their supportive environment during the pandemic led to feelings of isolation, loneliness, and stress, all risk factors for poor well-being. One well-being concern for older adults is the impact of the pandemic on cognitive health.

This study sheds light on cognitive health vulnerabilities by illustrating specific experiences among Middle Eastern/Arab immigrants. In particular, the references to symptoms of depression lead to a deeper understanding of how psychological distress is experienced in this group, and area in need of further research. Further, descriptions of communal interaction loss point to the centrality of relationships that extend beyond family interactions. Both depressive symptoms and social isolation increase the risks of cognitive health problems [37]. Importantly, participants described lingering changes in social connections following the lifting of the pandemic restrictions. Future studies should analyze the long-term effects of the breakdown in traditional support systems for older Middle Eastern/Arab American immigrant adults. This will help in assessing the needs of older immigrant adults and the potential for interventions to provide a supportive structure. 

This study sheds light on the ways in which multiple stressors were endured by an immigrant community. In so doing, its findings have implications for both research and policy including the need to incorporate ethnic factors to better understand health disparities and inform future policies to better meet needs when naturally occurring supportive systems are jeopardized.

### 4.2. Limitations

A potential limitation of this focus group includes recall bias linked to the recollection of the COVID-19 pandemic experiences. Recall bias may lead participants to overestimate or underestimate their experience during the COVID-19 pandemic, which would subsequently affect the study results [38]. Other limitations include the limited geographic locations. Though the Metro Detroit area is home to the largest and most visible concentration of Middle Eastern/Arab Americans in the United States, Middle Eastern/Arab immigrants in other parts of the country should be included in such investigations to increase the validity of the findings. 

## 5. Conclusions

The current qualitative study identified the ways in which older Middle Eastern/Arab American immigrants experienced challenges during the COVID-19 pandemic. Feelings of isolation, loneliness, and stress due to the lack of familial interactions and wider community support loss can negatively impact older adults’ cognitive well-being and mental health. Though interruptions in support systems were faced by all during the pandemic [5], this study unveiled unique meanings attributed to these hardships. Older Middle Eastern/Arab American immigrants rely on frequent contact with family and the wider community for support and well-being [21]. Notably, older Middle Eastern/Arab American immigrant adults emphasized the loss of valued support structures during the imposed social distancing measures. These changes led to heightened feelings of depression and ultimately may pose risks to their cognitive health. The meanings attributed to those experiences illustrate culturally embedded sources of stress unique to older Middle Eastern/Arab American immigrants. The results of this study emphasize the importance of how environmental context shaped by sociocultural factors impact cognitive aging and well-being.

## Figures and Tables

**Table 1 ijerph-20-05918-t001:** Participants’ demographic characteristics.

Characteristics	Frequency	Percentage (%)
Age (M ± SD)	73.6 ± 8.2	
Gender (female)	34	75.6
Marital Status		
Single	3	7
Married	15	34.4
Divorced	3	7.0
Separated	1	2.3
Widowed	21	48.8
Level of Education		
Less than high school	19	43.2
High school	13	29.5
College	12	27.3
Place of Birth		
Africa (Lebanese origin)	2	4.4
Iraq	13	28.9
Jordan	3	6.7
Lebanon	22	48.9
Palestine	4	8.9
Syria	1	2.2
Age Immigrated to the US		
Age 0–18	3	6.6
Age 19–49	23	51.1
Age 50+	19	42.2
Health Status ^a^		
Excellent	4	9.1
Fairly good	16	36.4
Average	21	47.7
Not very good	3	6.8
Average hours of care provided by family member (hours/day)	1.36 ± 5.05	

^a^ Health status was assessed by asking participants to reply to the question: “Taking all things into consideration, how would you rate your health at the present time? Would you say it’s excellent, fairly good, average or not very good”.

**Table 2 ijerph-20-05918-t002:** Semi-structured interview questions.

Section	Questions
**General Background and Household** **Relationships**	How would you describe your household’s general life and wellbeing before the quarantine began? Can you describe how the pandemic impacted your relationship to your family? Did the pandemic restrictions impact how often you saw your family members? ○ How does this relate to your pre-pandemic engagement with your family? ○ Were there some family members you continued to see? How did these changes in family engagement make you feel? How did this change your relationship with your family members? ○ Are your relationships between immediate and extended family members different now? Do you have family members who share responsibility in your health and daily well-being? ○ If yes, describe. What sort of care do they provide? ○ If no, please describe. Do you have family members who were unable to visit or help with your care due to COVID-19 stay at home orders or restrictions? ○ If yes, please describe. How did this impact you? ○ Probe: What are some of the things that you found challenging? Are there any other people who may assist with household / family needs or did so before the onset of coronavirus quarantine? What are some of the things you find challenging now?
**Medical Care**	How often did you seek medical services or care prior to the pandemic? Did that change during the pandemic? If so, what types of services did you no longer seek? Why? What difficulties did you face in using such social or medical services? Before the quarantine began, who would take you to your doctor’s appointments? Were your adult kids, brothers/sisters involved? Did that change during the pandemic? Did you miss some medical visits or care due to the pandemic? Was transportation an issue at all? How has quarantine affected your family’s ability to meet your general health needs? Did the way you receive care change during the pandemic? Was it virtual? When you did attend medical appointments, was the care the same? Did you have any difficulties while at the doctor’s office or medical center?
**Sociopolitical Background**	In what ways do your previous experiences (as immigrants or refugees) relate to experiences living under COVID restrictions? Does it relate to previous experiences either prior to arrival or as immigrants in the US? Do you see these experiences linked to politics in the US relating to Arabs and experiences of belonging? How does your memory and relationship to your past make you feel? Is this related to your health currently?

**Table 3 ijerph-20-05918-t003:** Themes identified from focus groups.

Name of the Code	Description	No. of Focus Groups Present in	References
**Fear**	Experiencing fear, worry, or anxiety associated with COVID-19	4	44
Fear		4	24
Scared or terrified		4	19
Terror		1	1
**Mental health**	Showing depressive symptoms or experiencing negative emotions	5	167
Depressive symptoms		4	29
Loneliness		5	48
Sadness		3	14
Stress		1	2
**Social relationships**	Discussing family connections and changes in relationships with immediate and extended family, as well as friends, during COVID-19	6	338
Immediate family		5	95
Family connections		6	151
Extended family relationships		5	92
**Total codes:**			549

Bolded words indicate coded themes identified across the focus groups.

## Data Availability

Data for this study is unavailable due to privacy and ethical restrictions. Data that is shared is in accordance with consent provided by the participants in the use of confidential data.

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
