# Peer review of "Self-Reported Cognitive Aging and Well-Being among Older Middle Eastern/Arab American Immigrants during the COVID-19 Pandemic"

_ijerph, 2023, doi:10.3390/ijerph20115918_

Round 1
Reviewer 1 Report
The paper is socially and scientifically relevant. Well written, concise, but dense. To enhance interest in readers, I suggest:
Line 15: Add that this State concentrates the majority of Arabic immigrants in US
Line 22: What meanings? To exemplify, even shortly, will gauge interest in researchers.
Line 26: “Family” and “Qualitative research” are vague keywords. Replace for something that relates to “older Middle Eastern/Arab American immigrants".
Line 75: This paragraph is describing better the aim of this study than that in the Abstract
Line 410: To be well linked to the paragraph beginning on Line 376, add something like “Difficulties to going back to previous patterns of social interactions were also mentioned during focal group debates, showing an important isue to be faced nowadays”.
Author Response
April 28, 2023
Ronica Rooks, PhD, MD; Joyce Weil, MD, Guest Editors
International Journal of Environmental Research and Public Health
Dear Dr. Rooks, Dr. Weil and Reviewer 1,
We would like to thank you and the reviewers for the constructive comments provided to improve this manuscript. This manuscript reflects edits as per the reviewer’s helpful suggestions (detailed below).
My coauthors and I hope that you find that our current manuscript has been strengthened and we look forward to your response. We have addressed comments (in bold) from Reviewer 1.

Reviewer 2 Report
This manuscript reports qualitative focus group data from a sample of older Middle Eastern adults residing in the Detroit area. The focus is on how these individuals were impacted by the COVID-19 pandemic.
I thought the manuscript was very well presented, and I especially enjoyed the direct quotes from the focus group participants. I just had a couple of relatively minor concerns.
1. While the themes described by the authors are certainly interesting, on the surface they seemed to be pretty similar to the experiences of a lot of people during COVID (e.g., fear, social isolation, etc.). Ideally, this study would have included a non-immigrant or younger sample but since that isn't the case, I would like to see a little more about how this group is unique.
2. For readers who are not familiar with the software used to analyze the data, I would provide at least a brief description of how it is used to analyze qualitative data.
3. I would have liked to see a little more information about the research participants. For example, how many had close family members nearby? How many of them were employed?
4. I'm also wondering how these findings would compare to research done before the COVID-19 pandemic? I would think that loneliness and social isolation were problems in aging immigrant adults prior to COVID.
5. Overall, though, very interesting study.
Author Response
April 28, 2023
Ronica Rooks, PhD, MD; Joyce Weil, MD, Guest Editors
International Journal of Environmental Research and Public Health
Dear Dr. Rooks, Dr. Weil and Reviewer 2,
We would like to thank you and the reviewers for the constructive comments provided to improve this manuscript. This manuscript reflects edits as per the reviewer’s helpful suggestions (detailed below).
My coauthors and I hope that you find that our current manuscript has been strengthened and we look forward to your response. We have addressed comments (in bold) from Reviewer 2 below.

Reviewer 3 Report
Congratulatiosn. The subject is of extreme importance, but I believe that adjustments should be made to make the article more interesting and easier to read.
Title
As the evaluation format was an interview where people should report their feelings in the face of questions asked by the interviewers, the authors should add the expression self-report in the title. Title suggestion: Self-report Cognitive Aging and Well-Being Among Older Middle Eastern/Arab American Immigrants During the COVID-19 Pandemic.
Introduction
The sentence "Accordingly, this study was conducted to investigate the impact 46 of COVID-19 on aging Middle Eastern/Arab immigrants in Michigan, home to the largest, 47 most visible concentration of Middle Eastern/Arab Americans in the US." it is in the midle of introduction and refers to the objectives of the study.
The sentence "By taking a qualitative approach, this study provides unique insights into the 74 lived experiences of older Middle Eastern/Arab American adults and brings to light the 75 concerns impacting their mental health and cognitive well-being." should be taken out of the introduction and placed perhaps in the discussion section.
Materials and Methods
How was health status assessed (Table 1)?
In Table 1, does the expression age of immigration refer to the age at which the person immigrated to the United States or the time the person was in the United States? It's not clear to me.
Results
The interviewees' narratives are very interesting, but make reading long and tiring. One idea would be to leave one or two narratives in the main text and the rest as a supplementary file.
Discussion
Discussion section must discuss the results and their implications. This is done in some parts, but there is repetition of objectives and part of the methods.
Conclusions
During the pandemic, older adults were forced to take extra precautions due to the increased risk they faced. The current qualitative study sought to evaluate the risks and concerns faced by older Middle Eastern/Arab American immigrants. Our findings suggest that our participants have some similar experiences faced by older adults during the pandemic. However, our study unveils unique concerns faced by Middle Eastern/Arab American older immigrants who rely on their social relationships and context for support and well-being. Feelings of isolation, loneliness, and stress due to the lack of familial interactions and wider community support can negatively impact older adults’ cognitive well-being and mental health. Older immigrant adults lost valued support structures as a result of the social distancing measures imposed. Importantly, participants described lingering changes in social connections following the pandemic restrictions. Future studies should analyze the long-term effects of the breakdown in traditional support systems for older Middle Eastern/Arab American immigrant adults. This will help in assessing the needs of older immigrant adults and the potential for interventions to provide a supportive structure.
In the conclusion paragraph, what is in yellow is not a conclusion and must be removed. The authors must remember that the conclusion must respond to the objectives of the study and is made according to the data collected from the study. Because of this, it is not recommended that bibliographical references be placed in the conclusion section.
Authors should rewrite the conclusion section.
References
In the main text, the citation format is wrong according to the journal's rules. Citations must appear before the period and between square brackets.
Author Response
April 28, 2023
Ronica Rooks, PhD, MD; Joyce Weil, MD, Guest Editors
International Journal of Environmental Research and Public Health
Dear Dr. Rooks, Dr. Weil and Reviewer 2,
We would like to thank you and the reviewers for the constructive comments provided to improve this manuscript. This manuscript reflects edits as per the reviewer’s helpful suggestions (detailed below).
My coauthors and I hope that you find that our current manuscript has been strengthened and we look forward to your response. We have addressed comments (in bold) from Reviewer 3 below.

Round 2
Reviewer 3 Report
The authors provided some changes, but not completely.
Accordingly, this study was conducted to investigate the impact of COVID-19 on aging Middle Eastern/Arab immigrants in Michigan, home to the largest, most visible concentration of Middle Eastern/Arab Americans in the US. This sentence should be rewritten or deleted as it seems to be the purpose of the study.
The results section is too long. You could leave just a few descriptions and the rest be placed in support material.
The conclusion should be rewrited.
Author Response
Dear Dr. Rooks, Dr. Weil, and Reviewer 3,
We would like to thank you and the reviewers for additional comments to improve this manuscript. This manuscript reflects edits as per the reviewer’s helpful suggestions (detailed below).
My coauthors and I hope that we addressed your concerns and that our current manuscript has been strengthened. Our manuscripts include changes in red from our first round of our reviews, with track changes to address the second round of comments from Reviewer 3.
REVIEWER 3
- Accordingly, this study was conducted to investigate the impact of COVID-19 on aging Middle Eastern/Arab immigrants in Michigan, home to the largest, most visible concentration of Middle Eastern/Arab Americans in the US. This sentence should be rewritten or deleted as it seems to be the purpose of the study.
Thank you for this suggestion. We have rewritten this sentence to provide greater clarity.
- The results section is too long. You could leave just a few descriptions and the rest be placed in support material.
We appreciate your suggestion. We have significantly removed sections of our description to meet your recommendation.
- The conclusion should be rewritten.
This point is well taken. We have revised the conclusion according to the reviewer’s suggestion. We hope this has provided greater clarity.
